# Effect of Genetic Variability in 20 Pharmacogenes on Concentrations of Tamoxifen and Its Metabolites

**DOI:** 10.3390/jpm11060507

**Published:** 2021-06-04

**Authors:** Yuanhuang Chen, Lauren A. Marcath, Finn Magnus Eliassen, Tone Hoel Lende, Havard Soiland, Gunnar Mellgren, Thomas Helland, Daniel Louis Hertz

**Affiliations:** 1Department of Clinical Pharmacy, University of Michigan College of Pharmacy, Ann Arbor, MI 48109-1065, USA; cyuanh@umich.edu (Y.C.); hellandt@med.umich.edu (T.H.); 2Department of Pharmacotherapy, Washington State University College of Pharmacy & Pharmaceutical Sciences, Spokane, WA 99202, USA; lauren.marcath@wsu.edu; 3Department of Breast and Endocrine Surgery, Stavanger University Hospital, P.O. Box 8100, 4068 Stavanger, Norway; finnmagnus@gmail.com (F.M.E.); tone.hoel.lende@sus.no (T.H.L.); 4Department of Clinical Science, University of Bergen, 5021 Bergen, Norway; hsoiland@gmail.com (H.S.); Gunnar.Mellgren@uib.no (G.M.); 5Hormone Laboratory, Department of Medical Biochemistry and Pharmacology, Haukeland University Hospital, 5021 Bergen, Norway

**Keywords:** pharmacogenetics, tamoxifen, endoxifen, active metabolites, *CYP2D6*, *CYP3A4*, *CYP2C9*, *SLCO1B1*, breast cancer

## Abstract

Background: Tamoxifen, as a treatment of estrogen receptor positive (ER+) breast cancer, is a weak anti-estrogen that requires metabolic activation to form metabolites with higher anti-estrogenic activity. Endoxifen is the most-studied active tamoxifen metabolite, and endoxifen concentrations are highly associated with *CYP2D6* activity. Associations of tamoxifen efficacy with measured or *CYP2D6*-predicted endoxifen concentrations have been inconclusive. Another active metabolite, 4-OHtam, and other, less active metabolites, Z-4′-endoxifen and Z-4′-OHtam, have also been reported to be associated with tamoxifen efficacy. Method: Genotype for 20 pharmacogenes was determined by VeriDose^®^ Core Panel and VeriDose^®^
*CYP2D6* CNV Panel, followed by translation to metabolic activity phenotype following standard activity scoring. Concentrations of tamoxifen and seven metabolites were measured by UPLC-MS/MS in serum samples collected from patients receiving 20 mg tamoxifen per day. Metabolic activity was tested for association with tamoxifen and its metabolites using linear regression with adjustment for upstream metabolites to identify genes associated with each step in the tamoxifen metabolism pathway. Results: A total of 187 patients with genetic and tamoxifen concentration data were included in the analysis. *CYP2D6* was the primary gene associated with the tamoxifen metabolism pathway, especially the conversion of tamoxifen to endoxifen. *CYP3A4* and *CYP2C9* were also responsible for the metabolism of tamoxifen. *CYP2C9* especially impacted the hydroxylation to 4-OHtam, and this involved the OATP1B1 (*SLCO1B1*) transporter. Conclusion: Multiple genes are involved in tamoxifen metabolism and multi-gene panels could be useful to predict active metabolite concentrations and guide tamoxifen dosing.

## 1. Introduction

Tamoxifen is a selective estrogen receptor modulator used in the adjuvant treatment of hormone receptor positive (HR+), i.e., estrogen receptor and/or progesterone receptor positive breast cancer. Tamoxifen use for 5–10 years significantly reduces the risk of recurrence and improves survival among patients with HR+ breast cancer; however, the 10-year recurrence rate is up to 30% [1]. Tamoxifen is a weak anti-estrogen that requires metabolic activation. The major route of tamoxifen metabolic activation occurs through the demethylation of tamoxifen to N-desmethyl-tamoxifen (NDtam) followed by 4-hydroxylation of NDtam to the active metabolite 4-hydroxy-N-desmethyl-tamoxifen (endoxifen). A second active metabolite, 4-hydroxy-tamoxifen (4OHtam), is formed by 4-hydroxylation of tamoxifen. Endoxifen and 4OHtam have ~100× higher affinity to the estrogen receptor compared to the parent drug tamoxifen [2]. Previous pre-clinical and pharmacological studies indicate that these two active metabolites may be responsible for the anti-estrogenic effect and efficacy of tamoxifen [3,4,5,6]. Although several previous studies found an association between endoxifen systemic concentrations and tamoxifen treatment outcomes, two studies also reported that low 4OHtam levels were associated with worse treatment outcome [3,4]. There are additional tamoxifen metabolites that may be partially responsible for tamoxifen’s anti-estrogenic activity, such as 4′-hydroxy-N-desmethyl-tamoxifen (Z-4′-endoxifen) and 4′-hydroxy-tamoxifen (Z-4′-OHtam). These 4-prime (4′) metabolites were reported to have ~10% of the anti-estrogenic activity of their Z-isomer counterparts [7]. However, detection and quantification of the 4′-metabolites requires chromatographic separation, as their molecular masses and fragmentation patterns are similar to other isomeric forms of endoxifen and 4OHtam. Several studies have used assays that did not account for this and, therefore, reported over-estimated concentrations of un-separated isoforms of endoxifen and 4OHtam [8]. Therefore, algorithms that include endoxifen and other active metabolites may be helpful to confirm the association of tamoxifen metabolite concentrations with tamoxifen treatment efficacy [9].

Endoxifen levels are highly variable among patients receiving 20 mg/day tamoxifen, and a major source of endoxifen variability is attributed to the activity of *CYP2D6*, the rate-limiting enzyme in the conversion of NDtam to endoxifen. The gene encoding *CYP2D6* is highly polymorphic, with over 100 allelic variants/haplotypes known, including several common variants associated with reduced or abolished *CYP2D6* phenotypic activity [10]. The *CYP2D6* genotype explains up to 50% of endoxifen variability and around 9% of 4OHtam variability [11]; thus, there are additional factors contributing to the formation of these two active metabolites [12]. Tamoxifen metabolism is complex and involves several other genes that may be predictive of endoxifen and 4OHtam concentrations, including CYP2Cs, CYP3As, SULTs, and UGTs [12]. The generation of active metabolites may require multiple metabolic steps for which genes responsible for upstream metabolism may affect concentrations of the downstream active metabolite. For example, endoxifen formation requires two steps: first, tamoxifen is demethylated to NDtam by one of several enzymes, and then it is hydroxylated to endoxifen, which is catalyzed by *CYP2D6* alone. Genes responsible for the generation of NDtam may, therefore, also be predictive of endoxifen formation. The same is true for Z-4′-endoxifen and nor-endoxifen, which are secondary and tertiary metabolites of tamoxifen, respectively. It was previously reported that *CYP2B6* and *CYP2D6* contribute to the formation of Z-4′-OHtam, but the genes associated with Z-4′-endoxifen remain unknown [11]. Few in-vivo studies on breast cancer patients using tamoxifen have investigated the genetic variables that affect non-endoxifen metabolites, as was recently reviewed by our group [12]. The objective of this study was to elucidate the effect of variation in 20 pharmacogenes on concentrations of tamoxifen and seven of its metabolites, measured by a gold-standard LCMSMS methodology.

## 2. Materials and Methods

### 2.1. Patient Cohort

Patients with breast cancer using adjuvant tamoxifen were recruited through the Prospective Breast Cancer Biobank, a prospective population-based biobank project collecting liquid biopsies and clinical data from breast cancer patients recruited at Haukeland and Stavanger University Hospitals in Norway [13]. Patients with estrogen receptor positive (ER+) breast cancer who were recommended tamoxifen as primary endocrine adjuvant treatment (*n* = 220) were included in a previous analysis investigating the associations of tamoxifen metabolite concentrations with treatment-related side effects [14]. From this population, patients with stored whole-blood samples for germline DNA purification were included in the current study. Data on *CYP2D6* inhibitor-use were obtained through the Norwegian Prescription database. All participants provided written informed consent before enrolling to the Prospective Breast Cancer Biobank and the biobank is approved by the Norwegian Regional Ethical Committee (2010/1957 and 2011/2161).

### 2.2. Sample Collection and Quantification of Tamoxifen and Metabolites

Serum samples drawn at least one month after tamoxifen initiation were collected in Vacuette™ serum tubes containing clot activator, coagulated for 30–60 min, centrifuged for 10 min at 2200 g and stored at −80 °C. EDTA whole-blood samples were collected on the day of surgery and stored at −80 °C. All patients were given instructions not to take anti-hormonal drugs on the morning of the blood draw.

Steady-state concentrations of tamoxifen, N-desmethyl-tamoxifen (NDtam), Z-4-hydroxy-tamoxifen (Z-4OHtam), 4′ isomer of Z-4OHtam (Z-4′-OHtam), Z-4-hydroxy-N-desmethyl-tamoxifen (Z-endoxifen), 4′ isomer of Z-endoxifen (Z-4’-endoxifen), tamoxifen-N-oxide (TamNoX), and N-N-didesmethyl-tamoxifen (NNDDtam) were assessed using a validated UPLC-MS/MS method for serum [14]. All metabolites and four deuterated internal standards (Tamoxifen-d5, 4OHNDtam-d5, Z-4OHtam-d5, NDtam-d5) were obtained commercially. Briefly, 20 μL serum samples were pre-processed using a Hamilton STAR pipetting robot (Bonaduz, Switzerland) and the resulting 80 μL supernatant was evaporated to dryness using nitrogen and reconstituted in 500 μL water:methanol (20:80, v:v). The samples were chromatographically separated by an Aquity UPLC system from Waters (Milford, MA, USA) using a Waters BEH Phenyl column before being subjected to atmospheric pressure photoionization and detected in positive ion mode using a Xevo TQ-S tandem mass spectrometer (Waters).

### 2.3. Germline DNA Isolation, Genotyping, and Activity Phenotype Prediction

Germline DNA was extracted from 400 uL whole blood at the ISO certified (ISO-9001) HUNT biobank, NTNU, Levanger, Norway, using the fully automated Chemagic Star workstation for nucleic acid extraction, Hamilton (Bonaduz, Switzerland). DNA was genotyped on the VeriDose^®^ Core Panel and VeriDose^®^
*CYP2D6* CNV Panel at Agena Bioscience (San Diego, CA, USA). The VeriDose^®^ Core Panel tested 68 SNPs/INDELs in twenty genes and five CNV assays. This panel was selected to provide broad coverage of the major functionally consequential genetic variability in important pharmacogenes, including polymorphic enzymes and transporters. The VeriDose^®^
*CYP2D6* CNV Panel includes seven regions in the *CYP2D6* gene and 22 CNV assays. This panel was selected to ensure that we had comprehensive genomic coverage of CYP2D6, which is known to be the most important pharmacogene in tamoxifen metabolism. All genetic information underwent appropriate quality control, including the assessment of sample call rate. Similar to our previous study [15], each allele was translated into the predicted activity (N = Normal, L = Low, H = High) based on published data and CPIC guidelines when available (Appendix A). Each patient’s diplotype was translated into predicted activity phenotypes (PM = Poor Metabolizer, IM = Intermediate Metabolizer, NM = Normal Metabolizer, RM = Rapid Metabolizer, UM = Ultra-rapid Metabolizer) for analysis (Appendix A), including *CYP2D6*, which was translated using consensus activity score designations from current CPIC guidelines [16].

### 2.4. Statistical Analysis and Data Analyses

Patients who had metabolite concentrations available and underwent successful genotyping were included in the analysis. Patients receiving a dose other than 20 mg/day (*n* = 1) or those with steady-state tamoxifen concentration <40 nM (*n* = 15), who were considered to be non-adherent to tamoxifen [5], were excluded from analyses. The data of tamoxifen and its seven metabolite concentrations were normally distributed. Linear regression was conducted to investigate all pair-wise correlations among tamoxifen metabolite concentrations. The association of genetic activity of *CYP2D6* [11] and all other genes with log-transformed tamoxifen metabolite concentrations were tested via univariate linear regression. Suggestive univariate genetic associations (*p* < 0.1) were included in a multivariable backward linear regression model with clinical covariates (age and body mass index (BMI)), which were selected based on their consistent associations with endoxifen and other metabolites [17]. To isolate the effect of genetics on the generation of the metabolite of interest, multivariable linear regression models also included all upstream metabolites. An uncorrected α = 0.05 (*p* < 0.05) was considered statistically significant for all variables in the multivariable analyses and only statistically significant results are reported. All statistical analyses were performed using IBM SPSS statistical software version 27 (SPSS, Inc., Chicago, IL, USA).

## 3. Results

### 3.1. Patient Data and Genetics

After excluding patients missing concentration or genetic data, patients receiving a daily tamoxifen dose other than 20 mg/day, and non-adherent patients, 187 patients were included in this analysis (Figure 1). Patient data, including demographics, tumor characteristics and treatment regimens, are reported in Table 1. The median age in this cohort was 48, 92.7% of the patients self-reported as White, and 87% were pre-menopausal. Genetic-defined metabolic phenotype frequencies for *CYP2D6* and all other genotyped genes are reported in Table 2.

### 3.2. Concentrations of Tamoxifen Metabolites

Steady-state concentrations of tamoxifen and all measured metabolites can be found in Table 1. The median steady state concentration of tamoxifen was 296.5 nM and the median concentrations of the active metabolites Z-endoxifen and Z-4OHtam were 28.3 nM and 5.0 nM, respectively, in line with other studies [3,6].

Pair-wise correlations between tamoxifen metabolite concentrations are reported in Table 3. Tamoxifen was significantly correlated with all downstream metabolites (all *p* < 0.01). Most metabolites were highly correlated with each other (*p* < 0.01), even those that were not in direct relation, or within, the same metabolic pathway.

### 3.3. Genetic Associations with Tamoxifen Metabolites

Multivariable linear regression was performed including genes with suggestive associations in univariate analysis, upstream metabolites, age and BMI (Table 4). Higher tamoxifen concentrations were associated with lower activity of *CYP1A2* and *CYP3A4*, which, when combined, explained 5.3% of tamoxifen concentrations. The major pathway responsible for tamoxifen metabolism is demethylation to NDtam. We found that NDtam was inversely associated with *CYP2D6* activity, which explained 11% of the variability in NDtam concentrations. Another route of tamoxifen metabolism is activation to Z-4OHtam. Higher Z-4OHtam concentrations were associated with higher *CYP2D6* and *CYP2C9* activity, and lower *VKORC1* and *SLCO1B1* activity, which, together, explained 14.2% of the Z-4OHtam concentrations.

Endoxifen is formed through two pathways, primarily by hydroxylation of NDtam, or via the minor route of demethylation of Z-4OHtam. In the univariate linear regression, *CYP1A2*, *CYP2D6*, *F5*, *SLCO1B1* and *VKORC1,* all had suggestive associations (*p* < 0.1, data not shown), but after backward linear regression, endoxifen concentrations were only associated with tamoxifen and the two intermediate metabolites, and increased *CYP2D6* activity (combined r^2^ = 0.79).

Next, the isomeride forms of Z-endoxifen and Z-4OHtam were studied. After backward linear regression, Z-4′-endoxifen concentrations were inversely associated with *CYP2D6* activity, whereas no genes were significantly associated with Z-4′-OHtam concentrations. The other tamoxifen metabolites, TamNoX and NNDDtam, were also measured. Higher TamNoX concentrations were found in patients with increased *CYP3A4* or decreased *SLCO1B1* activity. Higher NNDDTam concentrations were associated with increased *CYP3A4* and *CYP2D6* activity, and decreased *CYP2C19* activity.

## 4. Discussion

There is evidence that the pharmacological activity of tamoxifen is determined by its conversion to endoxifen [5,6,18,19], though this has not been consistently demonstrated [2,20]. These inconsistent findings may be partially due to the contribution of other active metabolites, such as 4OHtam [11,21,22,23], which has also been reported to affect treatment efficacy [3,4], and perhaps other, less active metabolites such as Z-4′-endoxifen and Z-4′-OHtam [7]. As is described in detail in our recent review [12], substantial work has been carried out to identify the genetic variables associated with endoxifen concentrations, but much less work has been conducted to identify the genetic variants that affect non-endoxifen metabolites. To investigate the genes associated with the tamoxifen metabolism pathway, which could facilitate further study of the efficacy of individualized tamoxifen treatment, we conducted an assessment of the effect of genetic variation in 20 pharmacogenes on steady-state concentrations of tamoxifen and its metabolites. This approach allowed us to generate a more complete understanding of the genes responsible for the metabolic conversion of tamoxifen to its downstream metabolites (Figure 2).

Tamoxifen concentrations were higher in patients with lower activity of *CYP3A4* and *CYP1A2*. *CYP3A4* is the primary enzyme responsible for the conversion of tamoxifen to its major metabolite, NDtam [11,25]. Our results agree with prior evidence that patients with reduced *CYP3A4* activity have higher steady-state tamoxifen concentrations, perhaps due to the reduced first-pass metabolism and increased bioavailability [26,27,28]. *CYP1A2* is responsible for demethylation of tamoxifen to NDtam [11], which could explain our finding, since lower *CYP1A2* activity leads to less conversion to NDtam. The conversion of tamoxifen to NDtam includes other enzymes, such as *CYP2D6* [11], which likely explains our finding that this gene was associated with the NDtam concentration. However, associations with other genes that have previously been reported to be associated with the formation of NDtam, including *CYP3A5*, *CYP2C9/19* [11,25,29], were not identified in our analysis. This may be due to differences between patient cohorts, including race, or the limited size of our cohort to detect genes with smaller effects.

An alternative pathway for tamoxifen metabolism is hydroxylation to Z-4OHtam, which is primarily catalyzed by CYP2D6 [23,30] with a minor role for CYP2C9 [11]. Our findings that higher CYP2D6 and CYP2C9 activity are associated with increased 4OHtam concentrations are consistent in direction and magnitude with previous studies [5,11,23]. We also found weak associations for SLCO1B1 and VKORC1. SLCO1B1 encodes the organic anion-transporter polypeptide 1B1 (OATP1B1), an influx transporter on the sinusoidal membrane of hepatocytes [31]. OATP1B1 transports tamoxifen and endoxifen [32], but our results found that decreased SLCO1B1 activity is associated with higher 4-OHtam levels. This may indicate that 4-OHtam is also transported by OATP1B1, though this has not been studied to our knowledge. VKORC1 encodes the catalytic subunit of the vitamin K epoxide reductase complex, which facilitates the activation of vitamin K. The association for VKORC1 is difficult to explain mechanistically and may be a false positive, due to the many association tests conducted without proper statistical correction. The final step in the metabolic pathway from tamoxifen to endoxifen is the hydroxylation of NDtam to endoxifen [29,30]. Our results agree with the extensive prior literature [12] that patients with greater CYP2D6 activity have higher steady-state endoxifen concentrations. Several studies have reported an association between CYP2C9/19 and endoxifen levels [12]; however, this association was not identified in our analyses. CYP2C9/19 is responsible for generation of Z-4OHtam and CYP2C19 is involved in the demethylation of Z-4OHtam to endoxifen. Since our regression analysis included Z-4OHtam levels the effect of CYP2C9/19 was likely removed. The adjustment of upstream metabolites could eliminate the upstream effect and isolate the genes that actually contribute to the metabolic conversion of interest.

We also investigated associations of pharmacogenes with the 4-prime Z-isomers of endoxifen (i.e., Z-4’-Endoxifen) and 4OHtam (i.e., Z-4’-OH-tam), which have a lower binding affinity to the estrogen receptor [7]. The genes responsible for the metabolism of NDtam to Z-4′-Endoxifen are unknown. Our analysis indicates an inverse relationship between *CYP2D6* activity and Z-4′-Endoxifen; however, this is likely due to the role of *CYP2D6* in generating the Z-isomers, thereby leaving more substrate (NDtam) available for Z-4′-endoxifen generation in patients with low *CYP2D6* activity. Our analysis did not identify any genes associated with Z-4′-OHtam after backward linear regression.

Finally, our study analyzed associations for two secondary metabolites with low or unknown anti-estrogenic effects: NNDDtam and TamNoX. *CYP3A4* and *CYP3A5* are believed to be responsible for the demethylation of NDtam to NNDDtam [11,24], which is consistent with our findings that *CYP3A4* is associated with NNDDtam concentrations. There may have been insufficient genetic heterogeneity in *CYP3A5* to detect this association, or *CYP3A5* may play a very minor role in this metabolic pathway. Interestingly, we also found stronger contributions from *CYP2D6* and *CYP2C19*, which are involved in the formation of the upstream metabolite NDtam [11,25,29] but not known to be involved in its downstream metabolism to NNDDtam. The enzymes previously thought to be responsible for the formation of TamNoX, *FMO1* and *FMO3* [24], were not included in our analysis. Our results indicated that *CYP3A4* and *SLCO1B1* were associated with TamNoX concentrations, but both explained a minimal portion of the variability after accounting for tamoxifen.

The effect of genetic variation on concentrations of tamoxifen and its metabolites may be useful to patients if the relationship between endoxifen, or some combination of its active metabolites, with tamoxifen efficacy is validated. The high correlation among the tamoxifen metabolites will make it challenging to identify the active metabolite or optimal combination, but that does not mean that a predictive biomarker could not be validated. Upon validation, genetic and clinical variables could be used to determine the optimal starting dose of tamoxifen [12], perhaps followed by a therapeutic drug monitoring approach to achieve the active metabolite concentration target [33], which could aid in improving the outcome of tamoxifen treatment such as the ten-year survival and recurrence rate. A recent study used panel genotyping to generate a population pharmacokinetic (PopPK) model that indicated *CYP2D6* IM and PM patients would require increases to 40 mg/day and 80 mg/day to achieve endoxifen concentrations similar to those seen in *CYP2D6* NM on 20 mg/day [34], which is consistent with the results of *CYP2D6*-guided tamoxifen dose escalation studies [35].

The strengths of our study included accurate quantification of tamoxifen and several of its major metabolites using a highly selective and sensitive UPLC-MS/MS assay [3,14]. Thus, the genotypic variants are only surrogate markers of the plasma concentration of the tamoxifen metabolites. These comprehensive metabolite data were combined with comprehensive genotyping for *CYP2D6*, including common variants and copy number variation, and 20 additional pharmacogenes. We also employed a novel approach of adjusting for upstream metabolites to isolate the effect of genetics on the metabolite of interest. However, this study had several limitations that should be considered. Some non-CYP genes previously reported to be associated with tamoxifen metabolism were not included in the pharmacogene panels used for genotyping, and therefore were not included in our analysis, including *SULT1A1* [36] and *UGTs* including *UGT1A1*, *UGT2B7* and *UGT2B15* [37,38,39]. Additionally, our cohort of ~200 patients from Norway lacks genetic heterogeneity for some genes and variants (e.g., *CYP3A5*3*) found primarily in non-Caucasians. Further research should be done in larger and more diverse cohorts to generate an algorithm that can integrate clinical and genetic data to precisely predict a patient’s steady-state concentrations of active tamoxifen metabolites.

## 5. Conclusions

This pharmacogenetic analysis confirmed that genetic variation in *CYP2D6* is the primary determinant of tamoxifen metabolism, with a secondary contribution from several other previously known genes including *CYP3A4* and *CYP2C9*. *SLCO1B1*, which encodes the OATP1B1 transporter, was identified as a novel gene that is associated with active tamoxifen metabolite concentrations. Upon validation that tamoxifen metabolites determine treatment efficacy, individualized treatment should be developed that integrates all relevant genetic and clinical information to select the appropriate starting doses. Future research is needed in larger and more diverse cohorts with more genetic and clinical information to develop a more precise algorithm for tamoxifen metabolism that can be used to improve treatment outcomes in patients with HR+ breast cancer in the absence of metabolite measurement.

## Figures and Tables

**Figure 1 jpm-11-00507-f001:**
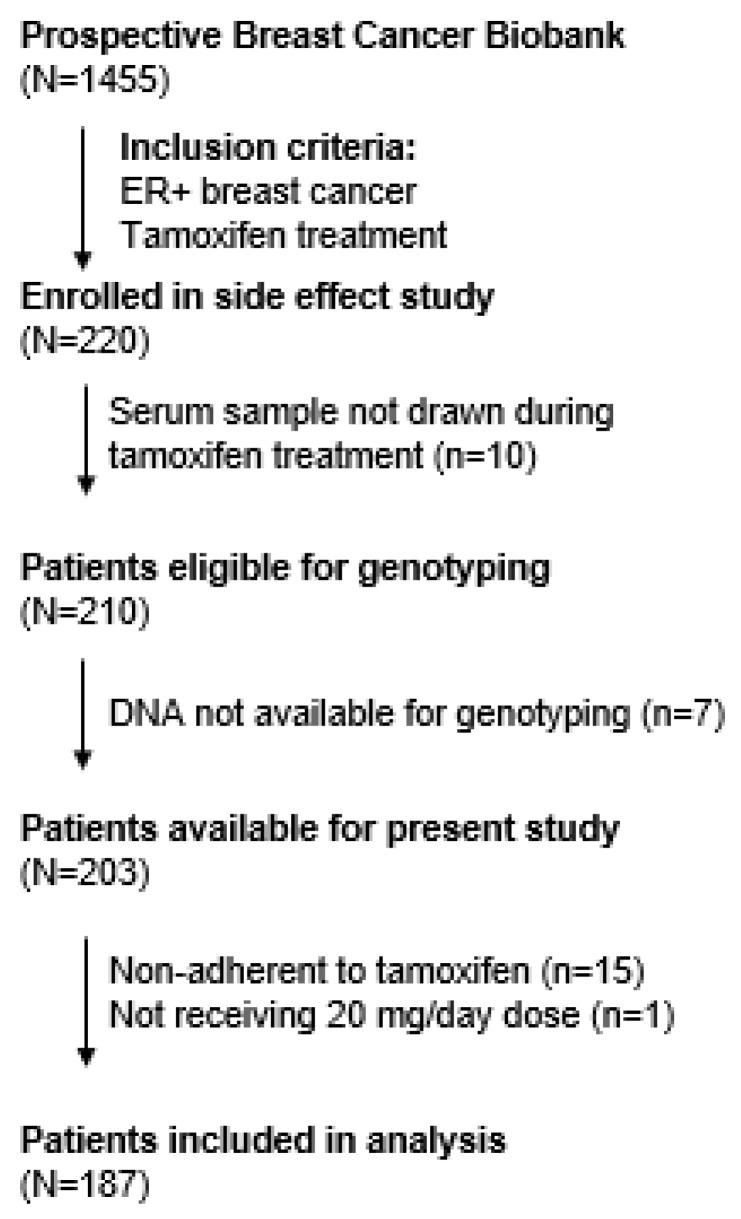
Patient flow from original cohort into present study. PROM = Patient Reported Outcomes. Non-adherence was determined by pharmacokinetic tamoxifen cut off 40 nM.

**Figure 2 jpm-11-00507-f002:**
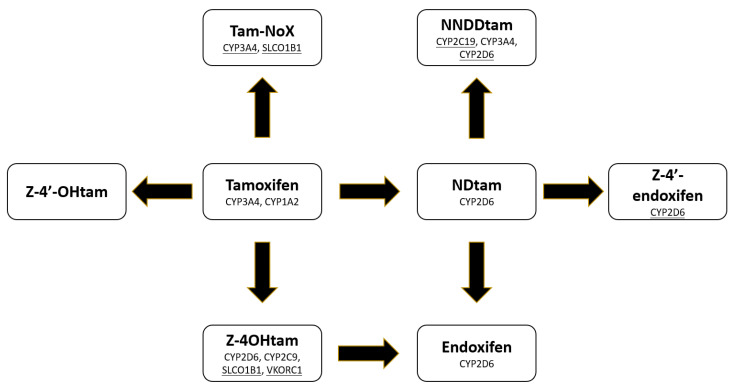
Tamoxifen metabolism pathway and genes associated with each step based on the results of this analysis. Genes associated with each metabolite are indicated within the figure. Normal text indicates our data confirm prior evidence of the gene’s contribution to this metabolic conversion [11,24], whereas underlined text indicates a finding that has not been previously described to our knowledge.

**Table 1 jpm-11-00507-t001:** Patient Demographics.

Clinical Variable	Level	Median (Range) or *n* (%)
Age	Years	48.00 (24.0–84.0)
Menopausal Status	Premenopausal	162 (86.6%)
	Postmenopausal	25 (13.4%)
Body mass index	kg/m^2^	24.6 (17.37–41.14)
Race	Asian	6 (3.2%)
	White	112 (59.9%)
	Other/Unknown	69 (36.9%)
Chemotherapy	No	19 (10.2%)
	Yes	168 (89.8%)
Time on tamoxifen	Months	12.0 (1.0—67.0)
Tamoxifen Metabolites	Tamoxifen	296.5 (56.17—898.66)
	NDtam	584.2450 (166.83—1551.44)
	Z-4OHtam	5.0170 (0.70—22.22)
	Z-endoxifen	28.3080 (4.45—96.00)
	Z-4′-endoxifen	22.9850 (6.70—62.97)
	Z-4′-Ohtam	7.3110 (1.72—23.00)
	TamNoX	18.6530 (3.60—70.42)
	NNDDtam	88.6410 (14.90—239.35)

Menopausal status is age determined (cut off 55 years). Race is determined according to U.S. Census Bureau.

**Table 2 jpm-11-00507-t002:** Genetically-defined Metabolic Activity Phenotypes.

Genes	Phenotypes (*n*, %)					
	PM	IM	NM	RM	UM	Missing
*CYP2D6*	13 (7.0%)	65 (34.8%)	98 (52.4%)		11 (5.9%)	
*CYP3A4*		22 (11.8%)	165 (88.2%)			
*CYP3A5*	155 (82.9%)	28 (15.0%)	1 (0.5%)			3 (1.6%)
*CYP2C19*	5 (2.7%)	51 (27.3%)	83 (44.4%)	42 (22.5%)	4 (2.1%)	2 (1.1%)
*CYP1A2*		4 (2.1%)	21 (11.2%)		162 (86.6%)	
*CYP2C9*	3 (1.6%)	63 (33.7%)	121 (64.7%)			
*CYP2B6*	11 (5.9%)	61 (32.6%)	115 (61.5%)			
*ABCB1*	30 (16.0%)	82 (43.9%)	75 (40.1%)			
*SULT4A1*	4 (2.1%)	35 (18.7%)	148 (79.1%)			
*APOE*	7 (3.7%)	7 (3.7%)	167 (89.3%)			6 (3.2%)
*COMT*	63 (33.7%)	88 (47.1%)	35 (18.7%)			1 (0.5%)
*DRD2*	10 (5.3%)	56 (29.9%)	120 (64.2%)			1 (0.5%)
*F2*		5 (2.7%)	182 (97.3%)			
*F5*		14 (7.5%)	173 (92.5%)			
*GLP1R*	39 (20.9%)	111 (59.4%)	34 (18.2%)			3 (1.6%)
*MTHFR*	14 (7.5%)	72 (38.5%)	101 (54.0%)			
*OPRM1*	5 (2.7%)	31 (16.6%)	151 (80.7%)			
*PNPLA5*	3 (1.6%)	40 (21.4%)	144 (77.0%)			
*SLCO1B1*	6 (3.2%)	53 (28.3%)	127 (67.9%)			1 (0.5%)
*VKORC1*	21 (11.2%)	90 (48.1%)	76 (40.6%)			

PM = Poor Metabolizer, IM = Intermediate Metabolizer, NM = Normal Metabolizer, RM = Rapid Metabolizer, UM = Ultra-rapid Metabolizer.

**Table 3 jpm-11-00507-t003:** Pearson Correlation Between Tamoxifen Metabolites.

		Tamoxifen	NDtam	Z4OH tam	Z-endoxifen	Z-4-prime-endoxifen	Z-4-prime-OH-tam	TamNoX	NNDD tam
Tamoxifen	Correlation		0.735	0.648	0.442	0.504	0.758	0.75	0.601
	*p*-value		<0.001	<0.001	<0.001	<0.001	<0.001	<0.001	<0.001
NDtam	Correlation			0.312	0.136	0.835	0.741	0.548	0.689
	*p*-value			<0.001	0.063	<0.001	<0.001	<0.001	<0.001
Z4OHtam	Correlation				0.839	0.203	0.606	0.511	0.351
	*p*-value				<0.001	0.005	<0.001	<0.001	<0.001
Z-endoxifen	Correlation					−0.029	0.346	0.385	0.385
	*p*-value					0.69	<0.001	<0.001	<0.001
Z-4-prime-endoxifen	Correlation						0.798	0.326	0.547
	*p*-value						<0.001	<0.001	<0.001
Z-4-prime-OH-tam	Correlation							0.501	0.549
	*p*-value							<0.001	<0.001
TamNoX	Correlation								0.41
	*p*-value								<0.001
NNDDtam	Correlation								
	*p*-value								

**Table 4 jpm-11-00507-t004:** Genetic and Clinical Factors Associated with Tamoxifen Metabolites.

Tamoxifen Metabolite	Variables	β Coefficient	Std. Error	*p*-Value	R^2^	
Tamoxifen	*CYP3A4*	−0.189	0.091	0.040	0.037	0.05
	*CYP1A2*	−0.155	0.072	0.033	0.016	
NDtam	Tamoxifen	0.716	0.039	<0.001	0.540	0.67
	*CYP2D6*	−0.174	0.022	<0.001	0.110	
	BMI	−0.333	0.101	0.001	0.020	
Z-4OHtam	Tamoxifen	0.710	0.052	<0.001	0.412	0.59
	*CYP2D6*	0.208	0.030	<0.001	0.111	
	*CYP2C9*	0.171	0.041	<0.001	0.015	
	*VKORC1*	−0.068	0.032	0.035	0.015	
	*SLCO1B1*	−0.085	0.039	0.032	0.001	
	BMI	−0.551	0.135	<0.001	0.036	
Z-Endoxifen	Tamoxifen	−0.349	0.110	0.002	0.003	0.79
	NDtam	0.310	0.101	0.002	0.017	
	4OHtam	1.038	0.072	<0.001	0.703	
	*CYP2D6*	0.361	0.037	<0.001	0.070	
Z-4’-Endoxifen	NDtam	0.989	0.067	<0.001	0.699	0.75
	Tamoxifen	−0.299	0.060	<0.001	0.029	
	*CYP2D6*	−0.046	0.023	0.048	0.010	
	BMI	−0.279	0.096	0.004	0.012	
Z-4’-OH-tam	Tamoxifen	0.680	0.041	<0.001	0.576	0.61
	BMI	−0.383	0.107	<0.001	0.034	
TamNoX	Tamoxifen	1.026	0.049	<0.001	0.563	0.59
	*SLCO1B1*	−0.162	0.037	<0.001	0.018	
	*CYP3A4*	0.191	0.061	0.002	0.012	
NNDDtam	NDtam	0.972	0.054	<0.001	0.490	0.62
	*CYP2D6*	0.183	0.029	<0.001	0.083	
	*CYP3A4*	0.243	0.061	<0.001	0.025	
	*CYP2C19*	−0.075	0.024	0.002	0.025	

## Data Availability

The data presented in this study may be available on reasonable request from the corresponding author. The data are not publicly available due to legal/privacy restrictions regarding patient data.

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
