# Peer review of "Effect of Genetic Variability in 20 Pharmacogenes on Concentrations of Tamoxifen and Its Metabolites"

_jpm, 2021, doi:10.3390/jpm11060507_

Round 1

Reviewer 1 Report

Review Report

The manuscript entitled “Effect of genetic variability in 20 pharmacogenes on concentrations of tamoxifen and its metabolites” reports the results of a pharmacogenetic analysis performed on 187 patients with breast cancer treated with 20 mg tamoxifen per day. The authors measured the concentrations of tamoxifen and 7 metabolites in serum samples using UPLC-MS/MS. Linear regression was performed to investigate all pair-wise correlations among tamoxifen metabolite concentrations, confirming that genetic variation in CYP2D6 is the primary determinant of tamoxifen metabolism, with a secondary contribution from several other previously known and novel genes.

  • Significance, strength and novelty of this study:

The study reports the results of an accurate quantification of tamoxifen and several of its major metabolites using UPLC-MS/MS assay and the comprehensive genotyping for CYP2D6, including common variants and copy number variation, and 20 additional pharmacogenes.

  • Limitations of the study have already been mentioned by the authors. Some non-CYP genes previously reported to be associated with tamoxifen metabolism were not included in the study. Their upcoming studies might include other genes and their products, offering a more detailed view of the allelic variants involved in tamoxifen metabolism.

Comments:

The study is well-designed, the results are clearly presented and the discussions highlight the key findings. Studying the effect of variation in 20 pharmacogenes on concentrations of tamoxifen and several of its metabolites represents a step towards the personalized treatment of breast cancer patients.

Minor comments:

  • Please explain the abbreviations.
  • Line 201: … were inversely associated with

Author Response

We thank the reviewer for their helpful suggestions. 

Note: Line numbers in this reviewer response document correspond with the continuous line numbering in the revised manuscript with tracked changes submitted as a supplementary document.

Reviewer #1:

The study is well-designed, the results are clearly presented, and the discussions highlight the key findings. Studying the effect of variation in 20 pharmacogenes on concentrations of tamoxifen and several of its metabolites represents a step towards the personalized treatment of breast cancer patients.

  1. Please explain the abbreviations.

Response: We have added the full description at the first instance of each of the abbreviations we used:

        “4’-hydroxy-N-desmethyl-tamoxifen (Z-4’-endoxifen)” (Page 2, Line 52-53)

        “4’-hydroxy-tamoxifen (Z-4’-OHtam)” (Page 2, Line 53)

  1. Line 201: … were inversely associated with …

Response: We have added “with” (Page 7, Line 237) to the manuscript, as suggested.

Reviewer 2 Report

This is a well-written manuscript. Some revisions would be helpful to support the non-familiar reader in his/her understanding, e. g. introduction of Z'-isomers, consistent use of terms (4'-... vs. 4-prime-...). The manuscript would also benefit if you provide some more information on the selection of the pharmacogenes. Please find more detailed comments in the attached document.

Author Response

We thank the reviewer for their helpful comments. 

Note: Line numbers in this reviewer response document correspond with the continuous line numbering in the revised manuscript with tracked changes submitted as a supplementary document.

This is a well-written manuscript. Some revisions would be helpful to support the non-familiar reader in his/her understanding, e. g. introduction of Z'-isomers, consistent use of terms (4'-... vs. 4-prime-...). The manuscript would also benefit if you provide some more information on the selection of the pharmacogenes. Please find more detailed comments in the attached document.

  1. (Line 32) SLCO1B1 is the OATP1B1 transporter and therefore not responsible for the metabolism of tamoxifen but for the distribution and/or elimination. Please modify the wording.

Response: We thank the reviewer for this observation and agree we should modify the wording about SLCO1B1. We have revised the sentence as suggested, “CYP3A4 and CYP2C9 were also responsible for metabolism of tamoxifen. CYP2C9 especially impacted the hydroxylation to 4-OHtam, and this involves the OATP1B1 (SLCO1B1) transporter.” (Page 1, Lines 32-33)

  1. (Line 34) to be checked if presented in the manuscript

Response: We have confirmed that we made several references to our results indicating that algorithms that account for multiple genes could be beneficial for patients receiving tamoxifen. For example, “Upon validation, genetic and clinical variables could be used to determine the optimal starting dose of tamoxifen [11],” (Page 9, Lines 336-337) and “individualized treatment should be developed that integrates all relevant genetic and clinical information to select appropriate starting doses.” (Page 10, lines 369-370). We have revised the sentence to better align with the text in the manuscript and clarify our intended meaning, “Conclusion: Multiple genes are involved in tamoxifen metabolism and multi-gene panels could be useful to predict active metabolite concentrations and guide tamoxifen dosing. (Page 1, lines 34-35)

  1. (Line 44) According to you statement below there are more than 'two' active metabolites. Please check and be more precise.

Response: We agree this seeming inconsistency could be confusing to the reader. We have revised the wording to more clearly state the metabolic routes, Tamoxifen is a weak anti-estrogen that requires metabolic activation. The major route of tamoxifen metabolic activation occurs through demethylation of tamoxifen to N-desmethyl-tamoxifen (NDtam) followed by 4-hydroxylation of NDtam to the active metabolite 4-hydroxy-N-desmethyl-tamoxifen (endoxifen). A second active metabolite, 4-hydroxy-tamoxifen (4OHtam), is formed by 4-hydroxylation of tamoxifen.” (Page 1, Line 43- Page 2, Line 55). We have also revised the sentence in question by replacing “these two active” with the two metabolites to which we are referring, “ Endoxifen and 4OHtam have ~100x higher affinity…” (Page 2, Line 55)

  1. (Line 51 – 53) I miss here a more detailed introduction of the Z-4' isomers. I am not sure if those isomers are always separated in the literature and regarded as separate metabolites.

Response: We agree that it would be beneficial to add more details about the Z-4’ isomers. As inferred by the reviewer, the Z-isomers are not always separated in the literature so relatively little is understood about their pharmacokinetics, pharmacology, or pharmacogenetics. We have added additional details to introduce these isomers:

There are additional tamoxifen metabolites that may be partially responsible for tamoxifen’s anti-estrogenic activity such as 4’-hydroxy-N-desmethyl-tamoxifen (Z-4’-endoxifen) and 4’-hydroxy-tamoxifen (Z-4’-OHtam). These 4-prime (4’) metabolites have been reported to have ~10% of the anti-estrogenic activity of their Z-isomer counterparts[7]. However, detection and quantification of the 4’-metabolites requires chromatographic separation as their molecular masses and fragmentation patterns are similar to other isomeric forms of endoxifen and 4OHtam. Several studies have used assays that did not account for this and therefore reported over-estimated concentrations of un-separated isoforms of endoxifen and 4OHtam [8].” (Page 2, Lines 61-69)      

  1. (Line 72- 74) Sentence is not clear. Please check and rephrase. Your are talking about 4-prime metabolites but mention only 4-OHtam. In publication [10] association between (z)-endoxifen and CYP2D6 is mentioned. What do you mean?

Response: We thank the reviewer for this observation and apologize for our unclear wording. We have clarified that publication 10 refers only to Z-4’-OHtam but the genes responsible for Z-4’-endoxifen are unknown, “It was previously reported that CYP2B6 and CYP2D6 contribute to the formation of  Z-4’-OHtam, but the genes associated with Z-4’-endoxifen remain unknown [11].” (Page 2, Lines 88-90)

  1. (Line 101) Please provide more information on this metabolite as you did for the others. For the non-familiar reader with this topic, it may be difficult to understand. e. g. Z-4OHtam isomer (Z-4'OHtam)

Response: We have specified that this refers to the 4’ isomer, as suggested, “4’ isomer of Z-4OHtam (Z-4’-OHtam)” (Page 3, Line 128)

  1. (Line 102) Same as for Z-4'OHtam

Response: We have added the suggested information, “4’ isomer of Z-endoxifen (Z-4´-endoxifen),” (Page 3, Line 129)

  1. (Line 143) I miss a statement that only significant results are presented as you tested several more genes than you describe in the results.

Response: We agree this is an important clarification and have added as suggested, “An uncorrected α=0.05 (p<0.05) was considered statistically significant for all variables in the multivariable analyses and only statistically significant results are reported” (Page 4, Lines 175-176)

  1. Do not use different terms for the same substance

Response: We agree with this suggestion and have revised the table accordingly.

  1. Same as above

Response: We agree with this suggestion and have made revision to the table.

  1. I find this figure confusing. You are presenting the associated CYPs and transporters close to the metabolism pathway arrows but according to my understanding you provide only the association between the drug or metabolite and the CYPs/transporters independent of the biotransformation pathway or why do you have sometimes 2 associations for one arrow? Suggest to include the association within the box.

Response: We greatly appreciate this recommendation, which we agree makes this figure clearer and more accurately represent our data. We have updated Figure 2 as recommended by the reviewer and converted it to black and white for easier transfer to gray scale for printing.

  1. (Line 269) I could neither find this expression in the cited publication nor I could find a lot in the internet on this. Suggest to use expression like isomer

Response: We apologize for the confusing terminology and have clarified the sentence by making it consistent with the terminology used throughout the manuscript, “We also investigated associations of pharmacogenes with the 4-prime Z-isomers of endoxifen (i.e., Z-4'-Endoxifen) and 4OHtam (i.e., Z- 4'-OH-tam),” (Page 9, Lines 310-311)

  1. There is no discussion or introduction how and why these genes were selected. Some are clearly associated with the biotransformation pathway of tamoxifen but for others it is not clear. Please provide some more information on your selection here or in the introduciton. There is also no discussion if the activity distribution of these genes is sufficient to identify any association.

Response: We agree this could have been more clearly explained and have added statements to the methods section to describe our rationale for selecting these panels,

“DNA was genotyped on the VeriDose® Core Panel and VeriDose® CYP2D6 CNV Panel at Agena Bioscience (San Diego, CA, USA). The VeriDose® Core Panel tested 68 SNPs/INDELs in 20 genes and 5 CNV assays. This panel was selected to give us broad coverage of the major functionally consequential genetic variability in important pharmacogenes, including polymorphic enzymes and transporters. The VeriDose® CYP2D6 CNV Panel includes 7 regions in the CYP2D6 gene and 22 CNV assays. This panel was selected to ensure that we had comprehensive genomic coverage of CYP2D6, which is known to be the most important pharmacogene in tamoxifen metabolism.” (Page 3, Lines 143-151)

We have also added a brief statement to the discussion to clarify that some relevant genes were not included in the analysis because they were not on the genotyping panels, “Some non-CYP genes previously reported to be associated with tamoxifen metabolism were not included on the pharmacogene panels used for genotyping, and therefore were not included in our analysis, including SULT1A1 [35] and UGTs including UGT1A1, UGT2B7 and UGT2B15 [37]–[39] (Page 10, Lines 354-358)

  1. (Line 321) If you mention a transporter 'metabolism' is not the right term. Maybe, you separate the sentences.

Response: We agree with the reviewer that SLCO1B1 should be a transporter and should not be stated in metabolism. We have removed SLCO1B1 from the sentence on metabolism and added an additional sentence for SLCO1B1: “SLCO1B1, which encodes the OATP1B1 transporter, was identified as a novel gene that is associated with active tamoxifen metabolite concentrations. (Page 10, Lines 366 - 368)
